# Environmental DNA from plastic and textile marine litter detects exotic and nuisance species nearby ports

**Aitor Ibabe**⊙*, **Fernando Rayón**⊙, **Jose Luis Martinez**, **Eva Garcia-Vazquez**

Department of Functional Biology, University of Oviedo. C/ Julian Claveria s/n. Oviedo, Spain

* ibabeaitor@gmail.com

## Abstract

Marine debris is currently a significant source of environmental and economic problems. Floating litter can be employed by marine organisms as a surface to attach to and use as spreading vector. Human activities are promoting the expansion of potentially harmful species into novel ecosystems, endangering autochthonous communities. In this project, more than 1,000 litter items were collected and classified from five beaches eastwards the port of Gijon, in Asturias, Spain. Next generation sequencing was employed to study biofouling communities attached to items of different materials. A dominance of DNA from Florideophyceae, Dinophyceae and Arthropoda was found, and four non-indigenous species (NIS) were identified. Results showed a clear preference of Florideophyceae and Bryozoa to attach on textile surfaces versus plastic ones. Considering that these taxa contain several highly invasive species described to date, these data emphasize the potential of textile marine debris as a vector for dispersal of NIS. Moreover, the closest beaches to the port contained a more similar biota profile than the farther ones, confirming that both plastic and textile marine litter can be vectors for species dispersal from ports.

**Data Availability Statement:** All relevant data are within the manuscript and its Supporting Information files.

## Introduction

Human activities have been triggering environmental changes all over the world since the beginning of intensive production methods. Human activities such as agriculture, fisheries, or industry, overexploit natural resources, and as a result, rates of species extinction are now 100 to 1000 times higher than prior to human influence [1]. A huge amount of the waste produced from this excessive human activity is ending up in the ocean, altering marine ecosystems. These materials are known as marine debris or marine litter. This problem has led to a difficult situation, not only for the conservation of marine ecosystems, but also for human health and economic activities. Plastic litter that is floating on the oceans is an important cause of mortality for many animals such as marine mammals, seabirds or turtles, either because they ingest it [2–3] or because they get entangled [4–6]. In addition, marine litter causes important economic losses in industries, such as fisheries, because of the time spent cleaning the debris from nets and net losses. As an example, marine plastics cost between $15 million and $17 million

**Funding:** This study was funded by the Spanish Ministry of Economy and Competitiveness, Grant CGL-2016-79209-R.

**Competing interests:** The authors have declared that no competing interests exist.

per year to the Scottish fishing industry [7]. Tourism can also suffer negative impacts due to the presence of marine litter on the coasts, which can affect the public perception of the quality of the surrounding environment and lead to a loss of income for this sector [8]. Besides, the degradation of plastic debris produces microplastics that can be transferred into the food chain and affect humans that consume them indirectly via contaminated marine food; this exposure to microplastics can result in chromosome alteration which can lead to infertility, obesity and cancer [9–10].

The role of marine debris as a dispersal vector of invasive organisms is of special concern [11]. Marine litter promotes the establishment and dispersal of NIS. It can provide a surface for colonizing species, facilitating their spread to new habitats [12]. Newly entered colonizers can get stablished and become alien invasive species (AIS) that alter the local ecosystem affecting the native organisms in several ways (competition, predation, habitat alteration, transmission of exotic diseases to local species) [13–15]. In addition to the impacts on local biodiversity, AIS have also severe impacts on the economy. In the United States, more than $138 billion are used every year to control new colonizers or to avoid infections of non-indigenous diseases [16]. Aquaculture industries are also affected by AIS that can alter the productivity, as in the case of *Undaria pinnatifida* which forms dense mats and obstructs light inhibiting shellfish growth [17], or *Carcinus maenas* which consumes native commercially important clams in Tasmania [18].

Identifying the biota that arrives in the local ecosystem is the only way to detect alien species and to control invasions. However, quite often invaders are spread in an early ontogenetic stage (e.g. eggs, larvae or algae propagules) and they are not visually identifiable, thus non-indigenous individuals may remain undetected until they are already adults and start reproducing and expanding [19–20]. Exhaustive monitoring is needed, but there is low probability of finding NIS because of their low density [21–22]. Identification based on organism morphology requires expert taxonomists specialized on the taxa to be analyzed, and often (especially in early development stages) identification cannot be done to a species level, limiting it to higher groups such as genus or family, which would not be useful for non-indigenous species identifications [23].

More recently, new techniques have been developed and species identification can be done based on sequencing and analyzing nucleic acids extracted from environmental samples [24], also called environmental DNA or RNA (eDNA, eRNA). Metabarcoding is a well-established method for the detection of NIS and for biosecurity applications [25–28]. In fact, techniques based on eDNA are advantageous when detecting species with low densities (such as exotic species at their arrival and before establishing), as very low DNA concentrations may be enough to find a species when the individuals are still very scarce and/or small [29–30].

Predicting invasions requires understanding the process of the invasion [31–32]; it is therefore crucial to understand how marine debris is spread, and to study the organisms with the capacity of attaching to these surfaces. Among some of the extensive work done on NIS transport via marine debris [33–40], some studies have shown the ability of biota to perform extreme transoceanic travels and survive over years attached to floating litter. For example, in 2011 a massive tsunami launched debris from the Japanese coast to Hawaii and North American shores. More than 280 living organisms native to Japan were documented attached to debris [41].

However, to our knowledge no research has examined the role of textile litter as a vector. We considered as textile litter, the disposed waste created during fiber and clothing production and the waste created by consumers use and disposal of textile products (including certain parts of sanitary pads that were classified as textile litter in this study). Ports are potential donors of both, marine litter and invasive species, therefore, studies on possible vectors

employed by biota inhabiting ports are needed, in order to predict potential future invasions and dispersions from these areas.

In this study, biota attached to litter items of different artificial materials were characterized by using next generation sequencing of DNA extracted from the biofilm, to analyze the composition of the communities inhabiting the marine debris. In order to assess a potential origin for biota found on litter, it was compared to the one growing on structures in the Port of Gijon (central south Bay of Biscay, Spain), where several NIS and AIS have been reported [42].

## Materials and methods

### Sampling

Beaches east of Gijon port were selected for our study for two reasons: (1) Gijon is a potential donor of marine invasive species; (2) in the winter, at the time of the study, dominant currents flow eastward along the coast [43], likely depositing debris from Gijon on beaches to the east. Therefore, five beaches located east of Gijon port were selected for litter sampling: Arbeyal, El Rinconin, Peñarrubia, Cagonera and La Ñora (Fig 1). No special access permits were needed as all samples were collected in public beaches.

From 13th to 17th of January 2017, litter items were collected from the five beaches. Sampling was carried out during the lowest diurnal tide (starting 2 hours before and ending 2 hours after) in order to increase the beach surface available to sample. The whole beach surfaces were sampled during these high-coefficient low tides and all litter pieces bigger than 5cm were taken. No transects nor quadrats were employed, as all the surface was analyzed and every litter piece was collected.

For a posterior characterization of the beaches, the litter was classified in situ in different types: sanitary pads, textiles, plastic bags, plastic bottles, expanded polystyrene (EPS) fragments, fishing gear, and others. After classification, only litter items containing visible biofilm were stored for posterior analyses. Samplings were taken in winter, at temperatures below 10°C and most of the collected litter items were discarded and recycled due to the lack of biofilm. A total of 16 items or item fragments from the beaches which were representative of the litter profile on each beach (approximately 0.25% of the total litter surface), were collected in sterile tubes and stored in ethanol for further biofilm sampling and extraction of eDNA.

### Taxonomy

For the names of the species we followed the taxonomic nomenclature from the World Register of Marine Species [44]. Regarding the status of the species detected visually and employing DNA, NIS and AIS were identified from the European Network of Invasive Alien Species Database NOBANIS [45]

### Environmental DNA extraction and metabarcoding

From all the litter items that were stored in ethanol, only 16 items belonging to different beaches and different types of material were selected for the eDNA extraction as the rest of the collected litter did not show any biofilm attached. Sterile swabs and gauzes were used to collect the attached biofilm from the litter (Fig 2) by scratching the surface. Sterile DNA/RNA free distilled water was used to rinse and clean the surface.

After the biofilm was recovered from the litter, the cotton extremes of the swabs were cut and collected with the gauzes in 15ml Falcon tubes with the water that was also employed to remove the biofilm. Then they were macerated for 2 minutes using a Stomacher 80 biomaster (Seward, UK) which was cleaned after each use with different samples to maintain sterility. A

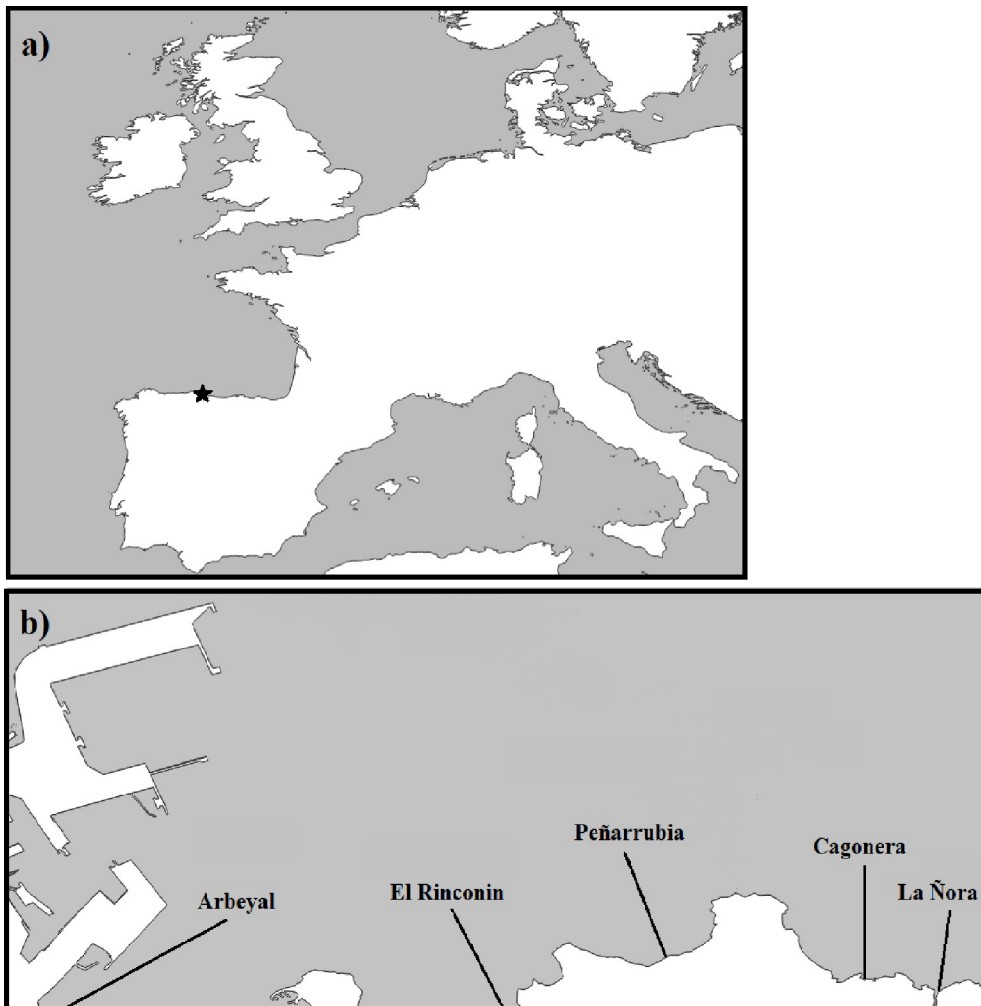

**Fig 1.** **a)** Location of the sampled area in the Northern coast of Spain. **b)** Location of the five sampled beaches eastwards the port of Gijon. Source: http://www.naturalearthdata.com/.

negative control was prepared for this whole procedure, by using sterile swab and gauze extremes and by suspending them into Sterile, DNA/RNA free distilled water. Once the Stomacher finished, excess liquid was squeezed from the swabs and gauzes and the suspension was pelleted by centrifugation (3000 x g 15 min) following the procedure reported by Pochon et al. (2015) [46]. The supernatant was discarded and then DNA was extracted from the pellet using an E.Z.N.A® Soil DNA Kit (Omega Bio-tek, USA) following the manufacturer's instructions.

The primers mICOIintF and jgHCO2198 [47] were employed to amplify a fragment of ≈300bp within the COI gene (miniCOI). Both primers were modified to include the specific sequences needed for Ion PGM libraries. A single common forward primer was used. Reverse primers were modified to include barcodes for each of the samples, so 16 different barcoded reverse primers were used. Each barcode has a known sequence to identify the samples after the whole process. Before sequencing, the quantity and quality of the DNA from PCR products was measured using Bioanalyzer (Agilent technologies). The PCR reactions were performed

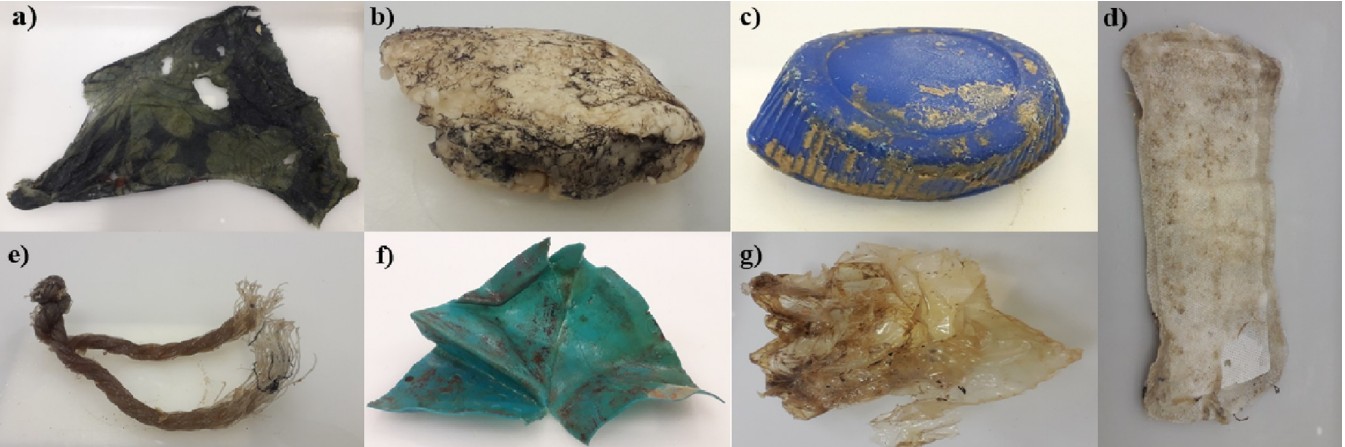

**Fig 2. Different sampled litter types showing biofilm that was scratched for eDNA extraction. a)** Fabric piece **b)** Expanded polystyrene **c)** Plastic bottle **d)** sanitary pad **e)** Fishing gear **f)** Plastic fragment **g)** Plastic bag.

using negative controls to monitor possible contamination. Thermocycling conditions were: 1x: 95˚C for 5 min; 35x: 95˚C for 1 min, 48˚C for 1 min and 72˚C for 1 min; 1x: 72˚C for 5 min and 4˚C on hold. The amplicons were analyzed directly in the platform Ion Torrent PGM (ThermoFisher Scientific, USA), in the Unit of DNA Analysis of the Scientific & Technical Services of the University of Oviedo.

## Bioinformatics pipeline for analysis of NGS data

Bioinformatics analyses were performed using QIIME1.9, an open-source bioinformatics pipeline [48]. Firstly, an initial screening was carried out in order to select reliable sequences, with a quality value > 20 and a length >200 bp. For taxonomic assignment, instead of using the whole GenBank as a reference, a specific database containing only eukaryotic COI sequences was generated with the script entrez.qiime (Chris Baker. ccmbaker@fas.harvard.edu. Pierce Lab, Department of Organismic and Evolutionary Biology, Harvard University). An initial assignment was made considering a minimum identity of 97% and an E-value of 1e-10 as these conditions were considered enough to obtain reliable species identification from COI barcodes [49]. In addition, assignments were also done employing minimum identity of 95% and E-value of 1e-50, to compare results. From the operational taxonomic unit (OTU) table obtained after the assignment, only marine and brackish taxa were retained for further statistical analysis. A subset of 50 sequences assigned to a species level from each parameter set were randomly taken from the OTU table. To double-check the reliability of the taxonomic identification of these sequences, they were assigned manually against GenBank using NCBI´s BLAST web browser (NCBI webpage, accessed July 2019).

## Statistical analyses

The statistical analysis was carried out with parametric or non-parametric tests done in PAST program [50] after checking normality in the dataset. For beach litter composition, the proportion of each type of debris was compared among beaches using non-parametric contingency Chi-square, confirmed from Monte Carlo procedure (n = 9999 permutations). The litter composition was compared between pairs of beaches using Euclidean distance, and the results visualized in a plot constructed from non-metric multidimensional scaling (nmMDS) analysis after checking stress and r2 in a Shepard plot.

The DNA dataset was analyzed with the following variables: the number of species of each taxon, the total number of species, the proportion of exotic species over the total number of species in each sample. Sequences assigned to terrestrial species and assignment artifacts (singletons and wrong species assignments due to the scarcity of reference sequences for certain taxa on NCBI database) were excluded from the analysis. Comparisons of the average number of species on plastics (as plastic bags, plastic bottles, buoys and expanded polystyrene) and textile objects (including sanitary pads and fabric pieces) were done using non-parametric Mann-Whitney tests. The community inferred from metabarcoding was compared between pairs of items using Gower's general similarity coefficient for presence-absence of each species, and nmMDS analysis was conducted as above. The same PAST software by Hammer et al. (2001) was employed.

## Results

### Beach litter

Beach surface area ranged from 2500 m2 in El Rinconin to 17500 m2 in La Ñora. A total of 1023 litter objects were found on the beaches; the corresponding densities were between 1.26 and 4.57 items/m2 in Arbeyal and Peñarrubia respectively (Table 1). Considering the litter surface area, it was between 2.46 cm2 of litter/m2 of beach in the cleanest Arbeyal to 18.6 cm2 of litter/m2 in the most littered Peñarrubia (Table 1). For litter surface La Ñora joined the group of more polluted beaches together with Peñarrubia and Rinconín, while for the number of items La Ñora beach was closer to the least polluted Arbeyal and Cagonera showing that few but big litter pieces were found on this beach.

The majority of litter (61.9%) was plastic, 33.9% was textile and only 43 objects (4.2%) were other materials. Textile items were mostly clothes but also sanitary pads (compresses) were included in this group although they are mainly composed by plastic. This is explained because eDNA for extraction was only taken from their textile part, which was the one with macroscopically observable biofilm. The five beaches were significantly different from each other for the type of litter ($\chi^2$ = 837.94; d. f. 40; p = 6.31x10-150; Monte Carlo p = 0.0001). For example, in Cagonera there were more textile items, while in La Ñora the predominant litter was small plastic pieces (Fig 3). Abandoned, lost or otherwise discarded fishing gears (plastic ALDFG) were found in all the beaches except in Arbeyal (the urban beach closer to Gijón port). None of this fishing gear showed any metallic parts (they were completely composed by plastic), in fact, metallic objects, like cans, were scarce in all beaches. They were found only on Rinconin beach (Fig 3).

**Table 1. Characteristics of the beaches sampled from the central south Bay of Biscay.** Beach surface in m$^2$. The litter density is given in surface as cm$^2$ of litter per m$^2$, and as litter items per m$^2$.

| | Arbeyal | Rinconín | Peñarrubia | Cagonera | La Ñora |
|---|---|---|---|---|---|
| Type of beach | Urban | Urban | Rural | Rural | Rural |
| Substrate | Sand | Sand | Pebble | Pebble | Sand |
| River | No | No | No | No | Yes |
| Beach surface area | 14000 | 2500 | 8250 | 10625 | 17500 |
| Latitude | 43.5445N | 43.5483N | 43.5518N | 43.5501N | 43.5471N |
| Longitude | 5.6934W | 5.6390W | 5.6237W | 5.6100W | 5.5897W |
| Litter density (cm$^2$ of liter /m$^2$ of beach) | 2.46 | 9.01 | 18.61 | 2.93 | 10.46 |
| Litter density (number of items/m$^2$ of beach) | 1.26 | 4.20 | 4.57 | 1.29 | 1.30 |

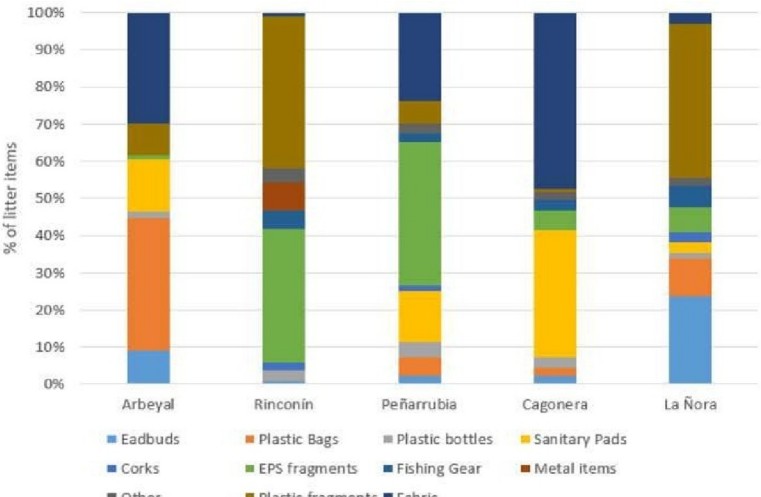

**Fig 3. Litter composition in the five beaches analysed in this study, presented as proportion of each type of item.**

The nmMDS based on Euclidean distances had stress of 0, r2 of 0.865 and 0.002 for the axis 1 and 2 respectively. Beaches were similarly connected in the minimum spanning tree regarding both, similarities for litter composition (Fig 4A), and similarities for biota composition (Fig 4B). Beaches that were richer in plastics (Rinconín and La Ñora) were quite proximate to each other but separate from those rich in textile (Arbeyal and Cagonera). Peñarrubia was isolated. The same connection was obtained for biota comparisons, with Peñarrubia beach again being the most different one.

## Biota on litter items identified with next generation sequencing

The surfaces sampled for biofilm and their composition are presented in Table 1. from all the litter items that were found (more than 1000), only the ones containing visible biofilm were stored for eDNA sequencing. In total they corresponded to 16 litter items from the different beaches, accounting for approximately 0.25% of the total litter surface. Only biofilms from 12 samples (from the initial 16 samples) provided DNA of quality to be successfully PCR-amplified and sequenced (Table 2). DNA sequences were not obtained from four expanded polystyrene pieces. For the 12 remaining biofilm samples, nine were from plastic objects and three were from textiles.

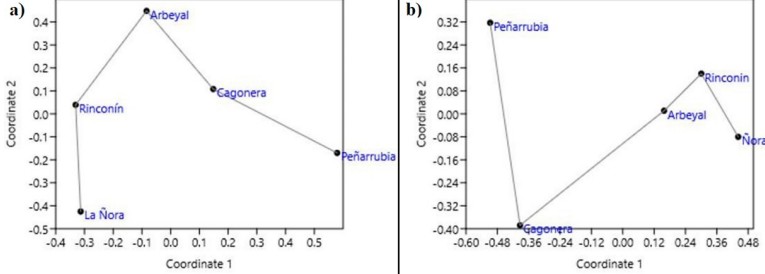

**Fig 4.** Non-metric multidimensional scaling analysis of the litter composition (**a**) and the litter biofouling biota identified from DNA (**b**), in the five analyzed beaches. Scatter plots constructed from pairwise Gower distances. The minimum spanning tree is presented.

**Table 2. Raw and filtered NGS results.** Litter surfaces used for biofilm analyses with the codification for each liter item (initial letter of the beach; P for plastic, T for textiles) concentration of eDNA as ng/µL, number of reads obtained before and after quality filters, and number of sequences assigned taxonomically after the final BLAST.

| Beach | Litter type | Material | Code | eDNA concentration (ng/µL) | Before quality filter (number of sequences) | After quality filter (number of sequences) | After BLAST (%97, $<E^{-50}$) (number of sequences) |
|---|---|---|---|---|---|---|---|
| La Ñora | Plastic Bottle | Plastic | Ñ-P1 | 2.66 | 57596 | 8476 | 969 |
| | Plastic Bag | Plastic | Ñ-P2 | 1.45 | 110931 | 33004 | 3417 |
| | Sanitary pad | Textile | Ñ-T1 | 2.09 | 8024 | 5915 | 97 |
| | Expanded polystyrene | Plastic | Ñ-P3 | - | - | - | - |
| Cagonera | Plastic fragment | Plastic | C-P4 | 1.88 | 1597 | 540 | 0 |
| | Fishing gear | Plastic | C-P5 | 2.22 | 99716 | 38098 | 1379 |
| | Sanitary pad | Textile | C-T1 | 1.98 | 93725 | 38877 | 3137 |
| | Expanded polystyrene | Plastic | C-P3 | - | - | - | - |
| Peñarrubia | Fabric piece | Textile | P-T2 | 1.79 | 14414 | 8501 | 1193 |
| | Expanded polystyrene | Plastic | P-P3 | 5.63 | 370143 | 91354 | 30241 |
| | Plastic Bottle | Plastic | P-P1 | 3.54 | 103369 | 24871 | 366 |
| Rinconín | Buoy | Plastic | R-P5 | 2.09 | 88324 | 19754 | 53 |
| | Expanded polystyrene | Plastic | R-P3 | - | - | - | - |
| Arbeyal | Plastic Bag | Plastic | A-P2 | 2.17 | 5332 | 707 | 81 |
| | Plastic fragment | Plastic | A-P4 | 3.45 | 21964 | 8027 | 131 |
| | Expanded polystyrene | Plastic | A-P3 | - | - | - | - |

The initial screening left 278 124 sequences (Table 2) that were useful for species assignments since they passed the quality filter (sequences >200bp and with a quality value >20). Although the same DNA amount of each sample library was employed for next generation sequencing, results were dissimilar, as for some samples much more sequences were obtained than from others (Table 2). The polystyrene piece from Peñarrubia (P-P3) was the sample from which more sequences were obtained (> 90000), while the plastic fragments from Cagonera (C-P4) provided the smallest number of sequences. After OTU assignment 66% of the sequences in P-P3 were lost (still remaining > 30000 sequences), and for the sample C-P4 none of the sequences assigned to a species with the employed BLAST criteria. So finally, biofilm communities were inferred from only 11 samples.

Species assignments made with a minimum identity of 90% and an e-value of 1e-10 retrieved many hits (S1 Table), but the reliability was too low because 82% of the manual individual BLAST did not assign the OTU to the same species. For >97% identity with the same e-value of 1e-10, despite much fewer significant hits retrieved, 45% of the sequences checked manually were assigned to a different species using manual BLAST. With a more stringent e-value of 1e-50 and 90% identity, the number of discrepancies between QIIME pipeline and the manual BLAST assignations was 22%. Finally, with an e-value of 1e-50 and 95% identity, all the putative species identified from QIIME coincided with those retrieved from manual BLAST. However, in order to increase the number of assignments to species level and not only to genus, a minimum identity of 97% was chosen, so that the final employed conditions were a minimum identity of 97% and e-value of 1e-50. Although 85% of the initial sequences were lost due to these highly stringent parameters the identifications obtained were very robust, as deduced from total coincidence with the manual BLAST.

In total, 122 species were identified from the eDNA present in the sampled litter. *Homo sapiens* and other non-marine species were detected, such as insects, mammals and freshwater organisms, but they were not considered for posterior analyses (S2 Table). Since we were working with debris like sanitary pads or plastic bottles, which are in contact with humans, we expected to obtain a lot of human sequences. Potential contamination with human DNA throughout the processing of samples can be discarded since no DNA amplification was detected in the negative controls. Insect species (specially Diptera) and big mammals like *Bos taurus* (cattle) and *Sus scrofa* (wild boar) were found in rural beaches like Cagonera where it is likely that runoffs had carried the eDNA from inland. In the case of insects, there is also the possibility that DNA belongs to eggs laid by adults on the debris such as it has been seen in the case of the marine insect *Halobates sericeus* that is known to lay eggs on marine debris and has been shown to benefit from the increase in marine debris in recent years [51].

Considering only marine and brackish taxa, 86 species classified into 17 major groups were identified from the analyzed samples. The putative taxa were not equally distributed in all the samples and beaches (Fig 5). In fact, some items showed a higher number of taxa than others. Sanitary pads from Cagonera (C-T1) provided more species (44 species) than the rest. On the other hand, biofilm from a plastic bag from Arbeyal (A-P2) only appeared to have a phaeophycean alga (*Petalonia fascia*).

The non-metric scaling analysis arranged the beaches from their fouling biota in an order similar to that found on the litter items (Fig 4A and 4B), with La Ñora, Rinconin and Arbeyal connected closer than Cagonera and finally Peñarrubia. This was connected with different types of biota found in biofilm from textile and from plastic litter. For example, more Florideophyceae (red algae) species were found on textile samples than on plastic ones (13 species were found on textile samples and only 9 on plastic; Fig 6A and 6B). For Dinophycea, more species were found on plastic litter (Fig 6A) than in textiles (Fig 6B). Only one species of Bangiophycea appeared, which was found on plastic from Rinconin beach. On the other hand, the two species of Echinodermata and the DNA of two species of Chordata (two Perciformes) that were found, only appeared on textile litter.

Textiles and plastics were compared for the number of species of each taxonomic group. Statistically significant differences between the two groups of litter items were found only for Bryozoa and Florideophyceae DNA, as a significantly higher number of species of these taxa occurred on textile items than on plastic ones (Mann-Whitney U = 0.5 with z = 2.764,

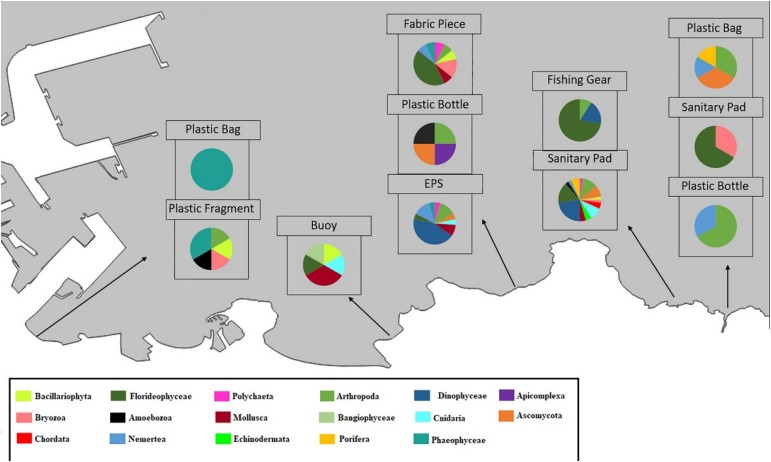

**Fig 5. Composition of the main taxa found on each litter sample.**

**Fig 6.  a)** Composition of biota occurring on plastic litter surfaces. **b)** Biota occurring on textile litter surfaces.

P = 0.006; and U = 3 with z = 2.062, P = 0.03, for Bryozoans and red algae respectively). In fact, for Bryozoans most plastic litter samples showed no species and only a single species was detected in one plastic piece (standard deviation = 0.333) while for most textile samples two Bryozoan species where found (standard deviation 0.577). On the other hand, most plastic samples showed no Florideophyceae species attached, although one sample recorded 8 different species (standard deviation = 2.645). Nevertheless, all textile samples had more than 4 different Florideophyceae species (standard deviation = 1) However, focusing only on the macrofauna profiles analyzed in previous studies in the region (i.e. number of species of the phyla Annelida, Arthropoda, Bryozoa, Chordata, Cnidaria, Echinodermata, Mollusca, Porifera published in Miralles et al. 2016), there was no significant difference between textile and plastic ($\chi^2$ = 7.885, 6 d.f., P = 0.247, and Fisher's exact test with P = 0.249 > 0.05, not significant).

DNA belonging to three exotic species was found in the dataset, including two species that are currently considered IAS (they are already stablished species that alter local ecosystems) in the study region: the brown alga *Sargassum muticum* (found on an EPS from Peñarrubia beach) and the signal crayfish *Pacifastacus leniusculus*. DNA assigned to *Pacifastacus leniusculus*, which is from brackish or fresh waters, was found on biofilm from a plastic bottle in La Ñora beach near the river (Table 3). The third species (Illex argentinus, NIS in the study area) was also detected in the EPS from Peñarrubia, a fragment of a box typically employed to transport fishing products. Thus, since the origin of DNA was likely from seafood catch remains and not true South American squid larvae, it was not taken into consideration.

Apart from these NIS and AIS, several native species were also found attached to the litter. Many of these species are considered potentially harmful because some strains can form toxic blooms (case of some dinoflagellate species), or produce diseases or allergies (Table 3). Some species do not cause any known toxicity or nuisance effects, but they are considered potentially dangerous in different places around the world where they are non-indigenous (NIS) or even invasive species (AIS). We detected Florideophycea species such as *Plocamium cartilagineum*, *Jania rubens* (aliens in the Mediterranean Sea), *Chondrus crispus* (alien in the United Kingdom) and *Gymnogongrus crenulatus* (alien in the Australian coast); Mollusca (*Mytilus edulis*; alien in the Black Sea); and Cnidaria (*Muggiaea atlantica*; invasion reports in Germany).

## Litter as a vector for species dispersal from Gijon port

For exploring the possibility of marine litter being a vector of dispersal from ports, the taxonomic profiles found in this study from beach litter were compared with published data from the port of Gijon. The comparison was done using the subset of marine macroscopic animal species only, because only macroscopic sessile animals were sampled in Miralles et al. (2016).

These samples were taken from three different sites in the port of Gijon; one near the port mouth, one in the inner section and another one half way between these two. Approximately

**Table 3. Non-indigenous and nuisance species which DNA was found attached to beached litter objects; Shaded in grey, species native from the study region that have been described as NIS or AIS elsewhere.**

| Taxon | Species | Reason for concern | Sample | Reference |
|---|---|---|---|---|
| Malacostraca | *Pacifastacus leniusculus* | AIS | Ñ-Plastic Bottle | [52] |
| Cephalopoda | *Illex argentinus* | NIS | P- Polystyrene | [53] |
| Phaeophyceae | *Sargassum muticum* | AIS | P- Polystyrene | [52] |
| Apicomplexa | *Isospora* sp. | Human parasite | P- Plastic Bottle | [54] |
| Ascomycota | *Cladosporium herbarum* | Asthmatic outbreaks and allergies | Ñ-Plastic Bag | [55] |
| | | | P- Polystyrene | [55] |
| | | | P- Polystyrene | [55] |
| Ascomycota | *Penicillium digitatum* | Rare pneumonia cases | C- Sanitary pad | [56] |
| Ascomycota | *Fusarium solani* | Infection of human cornea | C- Sanitary pad | [57] |
| Cnidaria | *Muggiaea atlantica* | AIS in Germany | C- Sanitary pad | [58] |
| Bivalvia | *Mytilus edulis* | NIS in the Black Sea | C- Sanitary pad | [59] |
| Dynophyceae | *Alexandrium catenella* | Paralytic shellfish poisoning | P- Polystyrene | [60] |
| Dynophyceae | *Karenia brevis* | Respiratory irritation | C- Sanitary pad | [60] |
| Dynophyceae | *Peridinium* sp. | Toxic blooms | P- Polystyrene | [60] |
| Dynophyceae | *Alexandrium ostenfeldii* | Paralytic shellfish poisoning | C- Sanitary pad | [60] |
| Dynophyceae | *Karlodinium* sp. | Toxic blooms | C- Sanitary pad | [60] |
| Dynophyceae | *Alexandrium minutum* | Toxic PSP blooms | C- Sanitary pad | [60] |
| | | | C-Fishing Gear | [60] |
| Dynophyceae | *Alexandrium* sp. | May produce toxic blooms | P- Polystyrene | [60] |
| Dynophyceae | *Azadinium poporum* | Azaspiracid shellfish poisoning | C- Sanitary pad | [60] |
| Dynophyceae | *Prorocentrum micans* | Shellfish killing blooms | P- Polystyrene | [61] |
| Dynophyceae | *Scrippsiella* sp. | May produce high density blooms | P- Polystyrene | [62] |
| Dynophyceae | *Alexandrium affine* | NIS in China, Ukraine, California | C-Fishing Gear | [15] |
| Florideophyceae | *Plocamium cartilagineum* | NIS in the Mediterranean Sea | C- Sanitary pad | [15] |
| Florideophyceae | *Ellisolandia elongata* | NIS in the Belgian coast | R-Buoy | [63] |
| Florideophyceae | *Jania rubens* | NIS in the Mediterranean Sea | P-Fabric piece | [64] |
| Florideophyceae | *Chondrus crispus* | NIS in the United Kingdom | C-Fishing Gear | [65] |
| Florideophyceae | *Gymnogongrus crenulatus* | NIS in the Australian coast | C-Fishing Gear | [66] |
| Phaeophyceae | *Leathesia marina* | NIS in the Mediterranean Sea | A-Plastic fragment | [67] |

200m$^2$ of artificial pot structures were sampled in each site, where visual inspections were done prior to sampling in order to detect phenotypically different organisms and to identify as many different species as possible, targeting all macroscopic biota inhabiting the port of Gijon. A total of 24 species were published in the port [42] (S3 Table) which were identified by visual and also genetic methods (barcoding of COI gene).

The number of shared species across taxonomic groups found in litter biofilm from beaches and in the port was four out of a total of 44 macroscopic animal species, corresponding to *Platynereis dumerilii* and *Syllis gracilis*(Polychaeta) the mussel *Mytilus edulis*, and the limpet *Patella vulgata*. For further analysis, the macrofaunal species fouling on litter (whatever litter type, since no significant differences were found between textile and plastic for macrofauna species profiles) were organized by proximity to the port, considering together the beaches located in the same bay, Arbeyal, Rinconin and Peñarrubia on one group, Cagonera and La Ñora on the other (see Fig 1).

The profile of the fouled macroscopic fauna of the port and the litter found on closer beaches was more similar to each other than the biota of the litter found on farther beaches (Fig 7). The macrofauna profile of Gijon port published by Miralles et al. (2016) was not

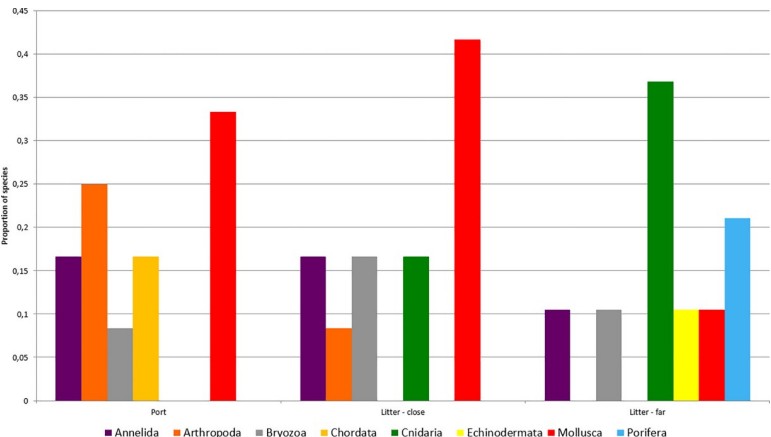

**Fig 7. Proportion of species of different animal groups fouling Gijon old port piers and litter from beaches near (Litter-close) or apart (Litter-far) the port.**

significantly different of that found on litter from the three closer beaches ($\chi^2$ = 7.797; 5 d.f.; p = 0.168, and Fisher's exact test with p = 0.193). In contrast, the taxonomic profile of the litter macrofauna found from the farther La Ñora and Cagonera beaches was highly significantly different from the port fauna reported by Miralles et al. (2016) ($\chi^2$ = 27.051;7 d.f.; p = 0.0003, and Fisher's exact test with p = 1.91x105).

## Discussion

Although based on a modest number of items, this study provided a number of results of importance in the field of environmental biosecurity. In the biofouling communities, as expected, some of the species detected from DNA are microscopic, such as dinophytes which are the most likely to survive attached to plastics or debris. For the macroscopic species, DNA was probably provided from free eDNA of macroscopic organisms, or from their microscopic life stages (early larvae, eggs, fragments). It is not possible to define if the detected species were alive organisms fouling marine litter, because DNA can persist for extended periods in the environment making discrimination of living versus dead organisms difficult [68]. Thus, we cannot confirm whether these results reflect an ability of the detected species to colonize new materials or not.

Environmental RNA (eRNA) is an increasingly employed molecular tool for metabarcoding based environmental characterization, and is being considered for biosecurity applications because it can be employed to distinguish living biodiversity [69–70]. However, some disadvantages related to eRNA are: overrepresentation of organisms with complex genomes and numerous copies of transcriptionally active marker genes [71], or different artifacts that can occur during RNA processing and PCR amplification [72]. Thus, further research is needed to achieve appropriate methodologies for metabarcoding based biodiversity characterizations.

Nevertheless, the presence of a wide variety of species was detected in this study, including non-animal taxa such as Florideophyceae, Phaeophyceae and Dinophyceae. These results are consistent with previous studies that confirm Cytochrome Oxidase I (COI) gene as an effective tool for the barcoding and identification of algae and Dinophyte species [73–76]. In fact, many of these algae sequences were randomly reviewed with individual BLAST and gave a robust assignment with >97% identity and high scores, so our study aligns with other authors who found COI to be a good tool to sequence red algae such as Florideophyceae [77].

A result to be highlighted was DNA of a significantly higher number of red algae and Bryozoa species found in textile debris than in plastic litter. Taking into account that both red algae [78–79], and Bryozoa [80–81] contain a high proportion of AIS, it seems that textile debris would have the potential to be a reservoir of potentially invasive species. Moreover, some of the species found in textiles are dangerous for public health because they may cause red tides (e.g. *Alexandrium minutum*) or produce infections (e.g. *Fusarium solani*, *Cladosporium herbarum*), thus the role of textile litter as a reservoir of species should be carefully taken into account. Previous studies have assessed marine plastic litter as a vector for nuisance species including human pathogens such as vibrio genus, or the dinoflagellates Ostreopsis so., Coplia sp. And *Alexandrium taylori*, known to form harmful algal blooms under favorable conditions [82–83]. Considering that 45% of the potential nuisance species that we detected were fouling textile litter, our data suggest that this type of litter could also be employed as a vector by these species, facilitating their spread into new habitats.

Fabric floatability is in principle lower than that of plastics, thus having lower dispersal capacity. However, on beaches with high litter accumulation the species accumulated in textile may pass on plastic items and eventually navigate offshore. This is why future studies should consider also other types of litter–in addition to plastics- in order to fully understand the role of marine litter as reservoir and dispersal vector of nuisance species.

Regarding the litter profile that was found in each beach, Cagonera showed a very high proportion of textile litter with many sanitary pads. This can be explained from a malfunctioning of the domestic wastewater treatment in the neighborhood. The neighbors were consulted about this and explained that the local wastewater treatment plant was temporarily closed and the toilets flushed directly to the beach. Thus, the large proportion of textile litter in that beach is likely not representative of the common beach state. Campaigns for not disposing this type of objects in toilets should be conducted in this area. On the other hand, Peñarrubia beach showed to be different to the rest of the beaches regarding litter composition and biota. This can be explained as Peñarrubia, unlike the other beaches, is not a sheltered beach, and it is also the only pebble beach located outside the city of Gijon (thus, it is not cleaned during winter). This could explain the high differentiation (regarding both, litter and biota) that this beach shows from the other ones.

Another interesting result was that the biota profile found on litter closer to the port was more similar to that of the port's macrofauna than litter collected further away. This can be considered a signal of species dispersal from the port using marine litter as a vector. In some cases, ports can be sinks as well as donors of species. However, the received species can be newly transferred to neighboring areas after arriving into a port [84]. In the case of the port of Gijon, marine currents flow eastwards in winter, so the port could be sink of litter coming from the west. However, this litter and species attached to it could be afterwards spread to the east from the port, which is why we consider it as the origin of the sampled litter items, as all the beaches are located eastwards the port. The macrofauna species found on litter were all native or cosmopolitan, suggesting that litter could not only transport alien species from Bay of Biscay ports [38] but also serve as a vector for the dispersal of native species, as it was found in Swedish waters [85].

Cnidaria were detected in great proportions on litter from far beaches and were not found on the port sampling; the detected species were millimetric polyps [86] that could adhere to floating debris (on adult or larvae stages). In fact, in the case of this type of hydroid fauna that was found, the sessile hydroid is more likely responsible for long range dispersal than the planktonic medusa stage [87].

The increase in the proportion of Cnidaria on distant beaches could be due to the fact that most of the species detected on the litter (such as *Clytia gracilis* or *Clytia paulensis)* are very

sensitive to disturbance [88] and these areas have a lower human impact than those closer to the city of Gijon. Similarly, it is also remarkable that species belonging to Porifera and Echinodermata phyla (classified as very sensitive to pollution) were only detected in distant beaches.

On the other hand, detected Annelida and Bryozoa species were all indifferent to pollution, and were found inside the port and in near and far beaches. These species have microscopic larval stages [89–90] that can adhere to litter items when floating as plankton and employ its surface in the adult phase (due to their size commonly smaller than 1mm) when they become benthic, leaving DNA traces that would be detected posteriorly. Some of the detected Mollusca were benthic species that also show a planktonic larval stage, that settle when the shell size is still smaller than 1mm [91] thus, early life stages could have employed marine litter to attach and spread.

Moreover, some of the native species that were found attached to marine litter (mainly Florideophyceae) are considered NIS or AIS in many other zones around the Globe, therefore, our results show that marine litter could be used as a spreading vector, facilitating exotic-species to reach and colonize new habitats.

Regarding local NIS and AIS, in the NGS results we detected DNA of several species, including an alga (*Sargassum muticum*), a cephalopod (*Illex argentinus*), and a freshwater crayfish (*Pacifastacus leniusculus*). NIS tend to be very difficult to identify in the initial phases of colonization, because their population size is normally small. This is an important issue because their eradication is easier in the first introduction stages when the population is not too big [92]. Sequences with low frequency occurrence, like those found in this study, should be taken into account, as they might be the key to anticipate or avoid possible future invasions. Following this approach, a deeper analysis is needed to correctly interpret the presence of DNA of exotic species on the particular litter objects analyzed in this study. The polystyrene piece sampled in Peñarrubia carried 15 DNA sequences identified as *Illex argentinus*. Individual BLASTs were made with some of the sequences belonging to *Illex argentinus* and confirmed that they were all correctly assigned. However, the Argentinean squid has no sessile life stages, and this species has never been detected in the Bay of Biscay. The origin of the polystyrene could explain this result; this material is employed in fishing vessels—polystyrene boxes are used to store the catch. Probably a polystyrene box used to store that squid ended on the sea and arrived in Peñarrubia beach, still containing remains of squid DNA in the biofilm. This data show that although eDNA is an important tool for an effective detection and identification of species, it does not guarantee detection of live organisms.

In contrast with *Illex argentinus*, the other two exotic species are considered invasive in Spanish waters. *Sargassum muticum* is a brown seaweed that has been already detected in Asturias [93] and alters local biodiversity triggering the decline of some native species such as *Gelidium spinosum* [94]. Our results suggest small propagules of this species could be transported attached to marine litter, using it as a spreading vector to colonize new environments. On the other hand, the presence of DNA of the freshwater signal crayfish (*Pacifastacus leniusculus*) in a plastic bottle (household origin) from La Ñora beach can be easily explained. This beach is in the estuary of River La Ñora, and eggs, larvae or naked DNA from freshwater organisms can arrive from the river, as rivers are conveyor belts of DNA diversity [95]. The species *Pacifastacus leniusculus* has been reported from River La Ñora [96] and our results are consistent with it, having a representation of the species living upstream.

On the technical side, next-generation sequencing was carried out with miniCOI amplicons in this study as COI is a largely studied gene and large amount of sequences are available [97]. Reference databases for the 18S gene are currently growing and the gene has been incorporated for example in BOLD (Barcoding of Life Diversity); however, the number of reference

sequences is still smaller for 18S gene [98]. For this reason, we based our study only on COI, that is one of the most represented DNA barcodes in public databases [99].

A problem for the use of Metabarcoding in biodiversity inventories is the unbalanced coverage of different taxonomic groups in current reference databases, especially in aquatic species [73]. For example, three sequences from a sanitary pad were assigned to *Squamamoeba japonica* (S1 Table) which is a deep-sea Pacific amoeba [100]. This could be an assignment artifact due to the scarcity of references because in July 2019 the only sea amoebas represented in GenBank with COI gene were of this species. It is possible that some DNA sequences of other marine amoebas were erroneously assigned to it. These assignment artifacts may happen not only with amoebas but with any other species that are not well defined on databases. So, in order to avoid this type of errors on future studies, the need of constructing global well referenced databases is remarkable.

## Conclusions and management recommendations

In this study, potentially dangerous species for ecosystem and for human health have been found employing DNA analysis of biofilm fouling litter objects. Textile objects, although likely less mobile than plastic ones, carried a significantly higher proportion of nuisance species. On the other hand, the macrofauna profile of litter objects found on beaches seemed to be associated with distance from the port, the closer the beach the more similar the macrofauna profile of litter. From the results obtained in this study, we consider that together with the general public concern about plastics and microplastics, more attention should be paid to textile litter. Preventing litter dispersal from ports is another important recommendation for avoiding exotic species spread.

## Supporting information

**S1 Table.**
(DOCX)

**S2 Table.**
(DOCX)

**S3 Table.**
(DOCX)

## Author Contributions

**Conceptualization:** Jose Luis Martinez, Eva Garcia-Vazquez.

**Formal analysis:** Jose Luis Martinez.

**Funding acquisition:** Eva Garcia-Vazquez.

**Investigation:** Aitor Ibabe.

**Methodology:** Aitor Ibabe, Fernando Rayón, Jose Luis Martinez.

**Project administration:** Eva Garcia-Vazquez.

**Resources:** Eva Garcia-Vazquez.

**Software:** Aitor Ibabe.

**Supervision:** Jose Luis Martinez.

**Validation:** Eva Garcia-Vazquez.

**Writing – original draft:** Aitor Ibabe.

**Writing – review & editing:** Jose Luis Martinez, Eva Garcia-Vazquez.

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
