## [Decision Letter · Decision Letter 0]

8 Nov 2019

PONE-D-19-21907

Plastic and textile marine litter as reservoirs for secondary species dispersal from ports

PLOS ONE

Dear Mr. Ibabe Arrieta,

Thank you for submitting your manuscript to PLOS ONE. After careful consideration, we feel that it has merit but does not fully meet PLOS ONE’s publication criteria as it currently stands. Therefore, we invite you to submit a revised version of the manuscript that addresses the points raised during the review process.

As you will see from their comment, the reviewers recognize merit, but also highlight some flaws in methodology, discussion and writing that need to be addressed. I agree with the criticisms raised and suggest that you modify your text accordingly. Once you submit your revision, it may undergo another round of revision.

We would appreciate receiving your revised manuscript by Dec 23 2019 11:59PM. To enhance the reproducibility of your results, we recommend that if applicable you deposit your laboratory protocols in protocols.io, where a protocol can be assigned its own identifier (DOI) such that it can be cited independently in the future. For instructions see: http://journals.plos.org/plosone/s/submission-guidelines#loc-laboratory-protocols

We look forward to receiving your revised manuscript.

Kind regards,

Raffaella Casotti

Academic Editor

PLOS ONE

Journal Requirements:

2. Our internal editors have looked over your manuscript and determined that it may be within the scope of our Plastics in the Environment Call for Papers. The Collection will encompass a diverse range of research articles to better understand various aspects of the effect of plastics in the environment. Additional information can be found on our announcement page: https://collections.plos.org/s/plastics-environment. If you would like your manuscript to be considered for this collection, please let us know in your cover letter and we will ensure that your paper is treated as if you were responding to this call. If you would prefer to remove your manuscript from collection consideration, please specify this in the cover letter.

Additional Editor Comments:

ear Authors, five authors reviewed your paper and found the information provided interesting. Yet, many expressed criticisms that need to be overcome. I suggest to address all criticisms raised, mopdify the text accordingly and resubmit. The revised manuscript might undergo another round of revision

Reviewers' comments:

Reviewer's Responses to Questions

**Comments to the Author**

1. Is the manuscript technically sound, and do the data support the conclusions?

Reviewer #1: Yes

Reviewer #2: Yes

Reviewer #3: Yes

Reviewer #4: Yes

Reviewer #5: No

2. Has the statistical analysis been performed appropriately and rigorously? 

Reviewer #1: Yes

Reviewer #2: Yes

Reviewer #3: Yes

Reviewer #4: Yes

Reviewer #5: No

3. Have the authors made all data underlying the findings in their manuscript fully available?

Reviewer #1: Yes

Reviewer #2: Yes

Reviewer #3: Yes

Reviewer #4: Yes

Reviewer #5: Yes

4. Is the manuscript presented in an intelligible fashion and written in standard English?

Reviewer #1: Yes

Reviewer #2: Yes

Reviewer #3: No

Reviewer #4: No

Reviewer #5: No

5. Review Comments to the Author

Reviewer #1: PLOS ONE –D-21907 manuscript

PLASTIC AND TEXTILE MARINE LITTER AS RESERVOIRS FOR SECONDARY SPECIES DISPERSAL FROM PORTS

Ibabe et al.

Plastic litter or marine debris is a huge problem for the marine ecosystem conservation, human health and economic activities. About the latter issue, plastic debris cause important economic losses in industries, such as fisheries, as time spent to cleaning the debris from nets and net losses. Tourism suffers for the presence of marine litter on the beaches leading to a loss of income for this sector.

Furthermore, plastic debris represents vector of transport of non indigenous species with impact on local biodiversity and the marine economy. It is difficult to investigate and classify taxonomically the NIS and invasive species on various vectors, especially when they are discharged from ports as donors to the local marine environment.

The study investigates the colonized microbiota on the debris, such as textile, plastics, fishing gears etc. using the metabarcoding method in order to compare the fauna attached on debris samples collected from donor port and debris samples collected from beaches.

The majority of litter was made of plastics (61.9%), textile (33.9%), and others (4.2%). The higher number of putative species retrieved on marine debris were from Bryozoa, Dinophyceae, Florideophyceae, Arthropoda. In particular, in textile debris Rhodhophyta and Bryozoa were retrieved representing a potential reservoir of invasive species. In fact, these taxa contain high proportion of invasive species. Textile objectives retained more harmful taxa than plastics. The macrofauna found on plastics collected from beaches was correlated to the macrofauna retrieved on debris from port.

The manuscript can be accepted after revisions.

Comments

-English revision of the text is needed.

-Dinophyceae has to be revised along the text.

-The authors used the primers targeting the COI region of metazoan (a taxon barcode for most animal taxa). These primers were targeting a barcode region that is specific for animal phyla (Leray et al. 2013). How this can fit with the mismatch and amplification of target mitochondrial regions of algae such as Dinophyceae and Rhodophyta? The authors should show the alignment of COI primer sequences with microalgae and macroalgae taxa COI sequences demonstrating the universal versatility of these COI primers designed for animal taxa.

-The retrieving of epi-flora and fauna on marine debris through the COI metabarcoding method did not demonstrate that eDNA were from alive individuals or dead/fragment organisms. As also the authors commented, a new approach of investigating the eDNA/eRNA metabarcode ratio could be more adequate in order to explain the effective risk that NIS or invasive biota can colonize new areas. This assumption can be better expressed in the text of the manuscript with an alert of what you are retrieving on debris fragments in the marine environment: dead or alive, fragment or entire individual/larvae, eggs, propaguli.

Instead, it is more likely that unicellular organisms such as marine microalgae can survive attached to plastics or debris.

- The NMDS is not clear; the authors can represent the distance correlation among the sites better.

Reviewer #2: I read with interest the manuscript "Plastic and textile marine litter ..."

I found the methodological approach scientifically proper, and reliable the results.

I suggest small intervention/corrections of the text directly on the manuscript.

I also would like to suggest Authors to better explain some their position, already from the introduction:

- Marine litter is not a problem of "nowadays": it exists from the beginning of the oceans. Probably Man inserted novelties (materials and /or quantities) in this framework, and really Authors probably speak about the only "Man derived litter"

- Ports could be considered sink as well as source areas for moving propagules. What is found on beaches, however, is more likely arrived material than ready-to be exported. Also in this case Authors should be careful in considerations and conclusions and I suggest a robust use of literature to justify such a point of view (they analyzed what arrived on the beaches, why they propose those materials are originated in the same area and ready to start?), or to re-examine the manuscript discussion from a different point of view

Reviewer #3: In this manuscript, authors use an eDNA and metabarcoding approach to characterize fouling organisms associated with biofilms on marine debris from beaches eastward of the port of Gijon in Spain. Researchers were able to detect several alien species using this method, underscoring the utility of this approach. Bryozoans and red algae dominated on textile surfaces compared to plastic debris, but macrofaunal composition did not vary between the two types of debris. Authors noted than many bryozoan and red algae species are invasives, highlighting the potential for textile debris to transfer these species into new areas. The researchers also observed that the macrofauna found on litter closer to the port was more similar to that of the port’s macrofauna than litter collected further away.

I commend the authors on tackling a timely problem with a molecular tool of emerging significance for documenting invasive species. The statistical data analysis and interpretation were relatively thorough, sound, and appropriate. The data support their conclusions. Authors appropriately addressed limitations of the study and offered good suggestions for future work. The main concern was the lack of detail regarding sample collection/selection. By providing this critical information, the manuscript will be strengthened. Also, an expanded discussion on the different types of organisms found near and far from the port, their life histories, and additional explanation for these observations would greatly improve the manuscript. Finally, additional review of grammar, vocabulary, and sentence structure would benefit the manuscript.

Suggested edits

Introduction

Line 14: Change “nowadays” to “currently” and “important” to “significant”

Lines 17-18: Change to “endangering autochthonous communities”

Line 20: Change “biofilm” to “biofilms”

Lines 43-44: Although likely, there are no known studies documenting health impacts on humans. If I am mistaken, please provide a citation for this statement.

Line 55: Add “a” between “provide” and “surface”

Lines 58-59: Commas are needed after “with” and “on”. Suggest removing “to” and replacing with “and”.

Line 64: Please further specify the impacts of Caluerpa taxifolia

Line 77: Suggest removing “Nowadays” and replacing with “More recently”

Lines 90-93: This sentence is not clear, please resolve

Line 95: Replace “on” with “of”

Line 97: Please explain how you derived the 1.5%

Materials and Methods

Change “Material” to “Materials”

Lines 108-110: As written, this sentence is unclear. Please clarify.

Please include more detail on sampling methods. Were samples collected along a transect? Using a quadrat? Randomly or haphazardly?

What was the outside temperature? Were samples placed on ice?

Line 135: Replace “employed” with “used” and replace “take out” with “sample” or “collect”

Line 139: Replace “take out” with “remove”

Line 165: Please specify which version of QIIME was used

Results

Line 224: Can you please explain what the term “sanitary pad” refers to in this study? In some places, this term is used to describe a woman’s menstrual product. Is that what it is describing here? I just want to clarify as these are typically made of a combination of cotton and plastic.

Lines 232-234: This sentence reads as out of place since the biota are discussed in the next section.

Lines 240-246: This paragraph might be better suited for the discussion.

Line 263: Please change the symbol to a percentage %

Line 276: Change “assignation” to “assignment,”

Lines 309-310: Suggest changing “; likely runoffs” to “, where it is likely that runoff”

Line 316: Replace “bigger amount” with “higher number”

Table 3: As a suggestion, the data may be presented in a more digestible format as a bar chart or pie chart paired with an inset of a map of the bay. I find myself having to keep referring to the map.

Figures 4A and 4B: Need percentages or species numbers labels associated with the pie charts

Line 351: Suggest replacing “nowadays” with “currently”

Line 358: Insert “a” after “from”

Line 379: Could you please provide a bit more detail regarding the published data from the port of Gijon (e.g. briefly describe what types of samples were collected and how they were analyzed.)

Discussion

Line 475: Suggest replacing “big” with “large”

Reviewer #4: This paper is important in adding to the work on biofilms on marine debris, and adding to the very small amount of work currently existing on the DNA sequencing of biofilms on marine debris. Their work about how animals spread port outward is important as we look at marine debris as a vector of invasive species and pathogens.

I have just a few revisions necessary:

There needs to be a clarification early on about what types of textiles these are, and how textiles fit in as a subset of marine debris. In reality, many textiles these days are almost fully synthetic, and so are themselves made of plastic. Especially sanitary pads are most likely majority plastic. If the researchers have access to FTIR spectroscopy, it would be a great addition to the paper to know if these clothing items were in fact 100% cotton, or wool, or mostly synthetic, and how that changed the biofilm composition.

Similarly, when fishing gear is discussed, it should be clarified if it was all plastic and synthetic fiber or if there was metal gear attached.

It is not the peer reviewer’s job to copyedit the manuscript, but there are some spots in this paper where the meaning is lost due to seemingly extra words, like lines 90-93, and some confusing phrasing. A careful proofreading is recommended.

Line 165-169: I do not believe it is proper to put the scripts used, but rather explain what that script did. Similar to 180-183, the actual website link does not need to be in the paper

Figure 3: Please explain more in the Results section. Why is Penarubbia apart?

Table 2: Explain the Code column in Table heading. What are units on last three columns?

Lines 308-311: There is also the marine insect Halobates that is known to lay its eggs on marine debris and has been shown to benefit from the increase in marine debris in recent years (Goldstein et al. 2012). If the DNA was from Halobates or their near relatives, it should not be excluded.

Table 4: a clarification between alien, invasive, and non-native species needs to be made early in the paper, or it needs to be clear that these are being used interchangeably. Are these of varying severities?

The other things in this table that are pathogenic need to be discussed more thoroughly.

Line 501-504: Why did you not use RNA if it would have been more effective?

Reviewer #5: While I appreciate the molecular work that went into this study, the results amount to a small research note, but not for PLOS ONE. Simply put, the study is far too limited -- a 5-day (!) sampling regime in one month (thus yielding no sense of temporal (especially seasonal) variation) based on only 12 analyzable samples. Critically, as the authors note (line 497 ff), viability of the detected organisms cannot be determined, critically limiting any interpretation (and thus my ranking of the challenges with statistical analyses).

As an aside, a lot of unnecessary verbiage introduces the article, which should start at lines 87-88 ("There are solid studies ...", but rather read, "There is an increasing number of studies...", not "solid"). All of the introductory material is unnecessary: there is no need to introduce readers to the basics of invasion science, marine litter, or DNA analyses, no more than one would introduce a reader to elementary descriptions of community or ecosystem ecology in a study on, for example, intertidal species). The paper should be edited by a native English speaker.

On a technical note, the "hit" of Squamamoeba japonica should be discarded out right -- it is not "difficult to explain" (as the authors write) -- it's a transparent artifact.

6. PLOS authors have the option to publish the peer review history of their article (what does this mean?). If published, this will include your full peer review and any attached files.

Reviewer #1: No

Reviewer #2: No

Reviewer #3: No

Reviewer #4: No

Reviewer #5: No

---

## [Author Response · Author response to Decision Letter 0]

28 Dec 2019

As our response to reviewers exceeds 20,000 characters we upload it as an attachment.

---

## [Editor Report · Decision Letter 1]

12 Mar 2020

PONE-D-19-21907R1

Plastic and textile marine litter as reservoirs for secondary species dispersal from ports

PLOS ONE

Dear Mr. Ibabe,

I note that recently you received word from the Academic Editor Dr. Casotti that your manuscript was accepted. However, when you submitted this manuscript, you submitted it for consideration in a special collection on Plastics in the Environment.

As such, after peer review for this manuscript was completed, it was sent to the Guest Editors who are overseeing the special collection for additional input. The collection editors have indicated that there are remaining points which need to be addressed related to the title and statistical analysis in this paper.

Specifically, the editors have indicated:

(1) The title should be adjusted to be more specific. For instance, the title should indicate what it is that is being dispersed (note the section of the title "secondary species dispersal". Dispersal of what in particular?) Additionally, noting that the technique picked up some species that are not being dispersed by plastic (i.e. cattle), the title should specify the e-DNA aspect of the analysis.

(2) The calculated means in the analysis should be provided alongside an estimation of error such as standard error or standard deviation. Similarly, discussion of estimation of error should occur alongside discussion of the means.

We would like to give you the opportunity to address these points. Therefore, we are returning this manuscript to your account so that you may address them.

Once you have addressed these points, we will provide the Academic Editor and the Collection Editors with the updated manuscript for a decision.

We would appreciate receiving your revised manuscript by Apr 26 2020 11:59PM. To enhance the reproducibility of your results, we recommend that if applicable you deposit your laboratory protocols in protocols.io, where a protocol can be assigned its own identifier (DOI) such that it can be cited independently in the future. For instructions see: http://journals.plos.org/plosone/s/submission-guidelines#loc-laboratory-protocols

We look forward to receiving your revised manuscript.

Kind regards,

Hanna Landenmark

Academic Editor

PLOS ONE
---

## [Author Response · Author response to Decision Letter 1]

30 Mar 2020

The title should be adjusted to be more specific. For instance, the title should indicate what it is that is being dispersed (note the section of the title "secondary species dispersal". Dispersal of what in particular?) Additionally, noting that the technique picked up some species that are not being dispersed by plastic (i.e. cattle), the title should specify the e-DNA aspect of the analysis.

Authors: OK, done. A new adjusted title was added: “Environmental DNA from plastic and textile marine litter detects exotic and nuisance species nearby ports.”

The calculated means in the analysis should be provided alongside an estimation of error such as standard error or standard deviation. Similarly, discussion of estimation of error should occur alongside discussion of the means.

Authors: OK, done. Values for standard deviation for the comparison of species presence over textile and plastic surfaces was added for Bryozoa and Florideophyceae groups, which show statistically significant differences.

---

## [Editor Report · Decision Letter 2]

21 May 2020

Environmental DNA from plastic and textile marine litter detects exotic and nuisance species nearby ports.

PONE-D-19-21907R2

Dear Dr. Ibabe,

We are pleased to inform you that your manuscript has been judged scientifically suitable for publication and will be formally accepted for publication once it complies with all outstanding technical requirements.

With kind regards,

Raffaella Casotti

Academic Editor

PLOS ONE
---

## [Editor Report · Acceptance letter]

4 Feb 2020

PONE-D-19-21907R1 

Plastic and textile marine litter as reservoirs for secondary species dispersal from ports. 

Dear Dr. Ibabe:

I am pleased to inform you that your manuscript has been deemed suitable for publication in PLOS ONE. Congratulations! Your manuscript is now with our production department. 

With kind regards,

on behalf of

Dr. Raffaella Casotti 

Academic Editor

PLOS ONE